# Long-Period Gratings and Microcavity In-Line Mach Zehnder Interferometers as Highly Sensitive Optical Fiber Platforms for Bacteria Sensing

**DOI:** 10.3390/s20133772

**Published:** 2020-07-05

**Authors:** Tinko Eftimov, Monika Janik, Marcin Koba, Mateusz Śmietana, Predrag Mikulic, Wojtek Bock

**Affiliations:** 1Centre de Recherche en Photonique, Université du Québec en Outaouais, 101 rue Saint-Jean-Bosco, Gatineau, QC J8X 3X7, Canada; predrag.mikulic@uqo.ca (P.M.); wojtek.bock@uqo.ca (W.B.); 2Institute of Microelectronics and Optoelectronics, Warsaw University of Technology, Koszykowa 75, 00-662 Warsaw, Poland or monika.janik@pg.edu.pl (M.J.); m.smietana@elka.pw.edu.pl (M.Ś.); 3Department of Metrology and Optoelectronics, Faculty of Electronics, Telecommunications and Informatics, Gdansk University of Technology, Narutowicza 11/12, 80-233 Gdansk, Poland; 4National Institute of Telecommunications, Szachowa 1, 04-894 Warsaw, Poland; mkoba@elka.pw.edu.pl

**Keywords:** optical fiber sensors, label-free biosensing, bacteria detection, long-period gratings, microcavity Mach-Zehnder interferometers

## Abstract

Selected optical fiber sensors offer extraordinary sensitivity to changes in external refractive (RI), which make them promising for label-free biosensing. In this work the most sensitive ones, namely long-period gratings working at (DTP-LPG) and micro-cavity in-line Mach-Zehnder interferometers (µIMZI) are discussed for application in bacteria sensing. We describe their working principles and RI sensitivity when operating in water environments, which is as high as 20,000 nm/RIU (Refractive index unit) for DTP-LPGs and 27,000 nm/RIU for µIMZIs. Special attention is paid to the methods to enhance the sensitivity by etching and nano-coatings. While the DTP-LPGs offer a greater interaction length and sensitivity to changes taking place at their surface, the µIMZIs are best suited for investigations of sub-nanoliter and picoliter volumes. The capabilities of both the platforms for bacteria sensing are presented and compared for strains of *Escherichia coli*, lipopolysaccharide *E. coli*, outer membrane proteins of *E. coli,* and *Staphylococcus aureus*. While DTP-LPGs have been more explored for bacteria detection in 10^2^–10^6^ Colony Forming Unit (CFU)/mL for *S. aureus* and 10^3^–10^9^ CFU/mL for *E. coli*, the µIMZIs reached 10^2^–10^8^ CFU/mL for *E. coli* and have a potential for becoming picoliter bacteria sensors.

## 1. Introduction

Bacteria have been a crucial target for rapid detection and identification in fields, such as food and water safety, medicine, or public health and security. Bacterial infectious diseases are one of the leading causes of deaths and hospitalizations worldwide [1]. For medical diagnostics and targeted treatment, a rapid identification of the bacterial strain is crucial. Moreover, during bacterial outbreaks, such examination can provide key information that can help to decrease the spread of the infection. Nowadays, conventional methods for bacteria-specific detection relay mostly on culture methods in selective media [2], immunological tests such as enzyme-linked immunosorbent assays (ELISAs) [3], or molecular techniques, such as real-time PCR [4]. Even though they are very effective, all of them are time-consuming, expensive, require extensive sample preparation, specialized equipment and technical staff to proceed with the tests.

Thus, there is an urgent need for more rapid, cost-effective, easy-to-use, sensitive tests, which could efficiently identify bacteria outside of the laboratory, or at the point of care. The future biosensors should be compact, operate at minimal sample volume, inexpensive, easy to work with, highly specific and sensitive. Moreover, they should be able to operate based on label-free methods, as these are relatively fast and do not need additional labeling/preparation of the tested sample.

Most of the optical fiber biosensors exploit the label-free target binding-induced changes, which alternate the sensor surface’s refractive index (RI), and the informationis then transduced to a detector and is typically identified as alteration of the output spectrum. For the purposes of biosensing, the changes in correspond to binding of a biological target, e.g., biomolecules or microorganisms, such as bacteria. Out of a great variety of optical fiber RI sensors these based on long-period gratings (LPGs) [5,6,7] and different versions of Mach-Zehnder interferometers [8] can be pointed out as the most sensitive.

The RI sensitivity of the LPG depends on RI range, and for commonly investigated LPGs is high around RI of fiber cladding, and fairly low around RI of water, where it is expected to be high when bacteria sensing is considered. For these LPGs, thin, high RI overlays may largely increase the sensitivity up to about 2000 nm/RIU [9] (RIU―Refractive Index Units). LPGs working in proximity of dispersion turning point (DTP) where double resonance (DR) effect can be observed, have a total sensitivity of both minima up to 800 nm/RIU around water and up to 10,800 nm/RIU in the range of 1.4436–1.4489 [10]. DTP-LPGs offer two resonance minima shifting towards opposite directions with RI increase. The DTP working conditions can be fine-tuned by etching the cladding [11]. DTP-LPGs are the most sensitive to RI changes, which makes them suitable for biosensing applications [12,13].

There is a wide selection of optical fiber Mach-Zehnder interferometers, which vary by fabrication technology, sensing length, type of interference, and sensitivity to RI [14]. A microcavity drilled across the fiber represents the most compact structure [15] referred to as micro in-line Mach-Zehnder Interferometer (µIMZIs). A Mach-Zehnder interferometer based on core cladding intermodal interference in photonic crystal fibers (PCF) has shown a sensitivity of 100–150 nm/RIU [16]. Other types of RI sensors are those based on mode coupling along with a two-core fiber exhibiting a sensitivity as highas 2100 nm/RIU [17], while RI sensors with asymmetrical tapered fiber based on evanescent field reaches as high as 3390–3914 nm/RIU [18]. Another approach is an interference in a single mode [19] and loss change in multimode [20] D-shaped fibers. In nanocoated D-shaped single mode sensitivity can reach up to 1566 nm/RIU in the 1.3–1.335 range [19]. Another structure is a micro-spiral upon a single-multi-single mode fiber section for which the RI sensitivity of 2114 nm/RIU in the 1.3373–1.4345 range can be achieved [21].

In the present paper, we analyze the capabilities of two most surrounding refractive index (SRI) sensitive structures namely DR-LPGs and µIMZIs, showing their applicability for bacteria sensing.

## 2. DTP-LPGs as Highly RI Sensitive Optical Fiber Platform

As outlined in the introduction LPGs, especially DTP-LPGs, exhibit high RI sensitivities and as such represent a desirable platform for bacteria and label-free biosensing in general. LPG is a periodic modulation of the RI introduced along a core of a single-mode optical fiber. The modulation causes a part of the power of the fundamental mode linearly polarized LP_01_ to be coupled to a higher order cladding mode LP_0p_ and they exchange power along the structure (Figure 1a). The propagation constants of the modes are defined as *β*_1_ = *n*_1_2*π/λ* and *β*_p_ = *n*_p_2*π*/*λ* for fundamental and high order modes, respectively, where *n*_1_ and *n_p_* are effective RI of core and cladding, respectively. Coupling from the core to the cladding modes occurs when the phase matching condition *δ**β* = 2*π*/*Λ* is met, where *Λ* is a period the modulation [22]. The resonance wavelength *λ_c_* at which the phase matching condition is fulfilled is found as in Equation (1).
(1)$$\lambda_{c}=(n_{1}-n_{p})\Lambda =\Delta n_{eff}\Lambda$$

Since higher order modes are strongly attenuated in the cladding, only the power remaining in the core mode is measured. At resonance conditions most of the power is coupled to the lossy higher order mode LP_0p_ so a minimum in the transmission is observed at *λ*_c_ (Figure 1a). The spectral position of this minimum depends on changes of surrounding RI, as well as temperature, axial strain, bending, and twisting. Bio-sensing applications of LPGs make use of the RI sensitivity which is very high around RI of fiber cladding (*n* = 1.445), but it is relatively low around RI of water (*n* = 1.333). Since bacteria stay alive in a water medium, the RI sensitivity of LPGs in proximity to that of water is expected. LPGs around DTP offer this property.

### 2.1. LPGs around the DTP

As seen in Equation (1) the resonance wavelength is proportional to the grating period. However, the effective RI difference Δ*n_eff_* is not constant with respect to wavelength *λ*. Because of Δ*n_eff_* dispersion, the grating period and the resonance wavelength for a given higher order cladding mode are related in a way as shown in the lower part of Figure 1b. For a given LP_0p_ mode at a period *Λ*_0_there is only one minimum. This marks the so called dispersion turning point (DTP). For smaller values of the period (*Λ* < *Λ*_0_) there are two resonance wavelengths: a lower *λ_L_* and a higher *λ_H_* (*λ_A,_*_L_ < *λ_A,H_*), which means that there are two different values of Δ*n_eff_* for each resonance wavelength, namely Δ*n_L_* < Δ*n_H_*. As seen from Figure 2b, as the period *Λ* decreases, *λ_L_* decreases while *λ_H_* increases. On the other side, once the grating is fabricated *Λ* is constant and if for some reason Δ*n_eff_* changes, then so do the two resonance wavelengths accordingly. This occurs if for example strain, temperature, or surrounding RI changes are introduced upon the grating.

Since DTP-LPGs are the most sensitive to RI, they are fabricated in a way as to be just before DTP when surrounded by air and to appear in water with a minimum at *λ*_0_. In such a manner, any change of RI above that of water will be detected as an increase of losses or a split in the spectrum with minima at wavelengths *λ_L_* and *λ_H_* going apart. The LPG thus becomes a double-resonance grating. The total wavelength shift between the two resonances will be then defined as in Equation (2).
(2)$$\Delta \lambda =\lambda_{H}-\lambda_{L}$$

In this paper, we make use of this effect to obtain maximum sensitivity of the LPG around DTP. The responses to RI of two DTP-LPGs are presented below. The first with a period of *Λ* = 135.6 µm at a center wavelength *λ*_c_ = 770 nm was UV written in a Fibercore SM600 single mode fiber with a cut-off wavelength in the 500–600 nm range. The second with a period of *Λ* = 226.8 µm at a center wavelength *λ*_c_ = 1583.4 nm was UV written in a Corning SMF-28 (cut-off at 1260 nm). The UV laser was a Krypton fluoride Lumonics KrF excimer laser. Both gratings fabricated at the labs of the Centre de Recherche en Photonique-UQO were etched using hydrofluoric HF acid to fine-tune them around the turning point in a way as to observe it in water immersion.

For the DTP-LPGs, two types of spectral changes are observed. The first (I) are the resonance wavelength shifts in opposite directions of *λ*_L_ and *λ*_H_ and hence the increase of Δ*λ* with RI. The second (II) is the increase of the power level (or loss decrease) Δ*I_DTP_* at *λ_DTP_*. In both the Δ*λ* vs. RI and the Δ*I_DTP_* vs. RI dependences, we note two ranges: the first one is for RI from 1.33 to 1.34 offering higher sensitivity when compared to the second range for RI > 1.34 exhibiting a lower sensitivity.

The Vis DTP-LPG around RI of water RI exhibits sensitivity of about 11,000 nm/RIU which falls to about 690 nm/RIU away from it (Figure 3a), while the IR DTP-LPG offers a sensitivity of about 6300 nm/RIU around RI of water and about 2700 nm/RIU away from it (Figure 4a). Similarly, in terms of change in power at fixed wavelength, the Vis DTP-LPG shows higher RI sensitivity of about 970 dB/RIU for RI below 1.34 that is reduced to 124 dB/RIU beyond that (Figure 3b).

The IR DTP-LPG has the same sensitivity in the whole RI range reaching about 317 dB/RIU (Figure 4b). In both the DTP cases the sensitivities around RI of water is much high as it is expected for biosensing applications.

### 2.2. Fabrication of LPGs

#### 2.2.1. Fabrication Technologies

LPGs can be arc-induced using a fiber splicer [23], fabricated using UV pulsed [24], femtosecond laser or CO_2_ laser. Comparative analysis shows that UV-written and arc-induced LPGs may offer similar sensitivities [25]. Out of the mentioned above methods, the UV laser method uses amplitude masks while the others are point-by-point writing methods. The arc-induced method is the simplest and the LPGs spectrum is stable in time and temperature. However, the minimum periods of the gratings are limited by the electric arc dimensions to about 180 µm. Moreover, the repeatability of their parameters is not as good as of the LPGs written with amplitude masks and UV lasers. Amplitute mask and point-by-point laser written LPGs are not limited in period length, but need additional post-fabrication annealing to stabilize the grating parameters in the long term. All point-by-point writing methods allow greater flexibility in medication of the grating periodicity or coupling strength along the structure. The described biosensors were fabricated using KrF excimer laser in combination with an amplitude mask and a subsequent etching.

#### 2.2.2. Fine-Tuning Approaches

The fine-tuning of the LPGs includes, basically, the tailoring of the sensitivity and the tuning of the LPG to a definite initial spectral distribution close to the DTP as shown above. This is basically done by cladding etching and nanocoating with, e.g., Al_2_O_3_ [26,27]. Sensitivities as high as 20,000 nm/RIU around RI of water have been achieved and reported [26]. The deposition of Al_2_O_3_ has been found also to cause center wavelength shifts [27]. Due to this, the sensitivity and the center wavelength of the DTP-LPG can be fine-tuned. However, since DTP-LPGs are highly sensitive not only to RI, but also to strain, temperature twists and bends, experimental difficulties with noise reduction arises in measurements with these gratings, which can be overcome by isolating the sensing grating from unwanted stimuli [28].

#### 2.2.3. Temperature Stability

Since DTP-LPGs are strongly temperature dependent, to ensure temperature stability LPGs are placed on temperature stabilized conditions or in glass tubes to eliminate temperature fluctuations caused by air flows. A more sophisticated method is based on a cascaded structure of two DTP-LPGs separated by a fiber section, which ensures a temperature stability of −0.45 pm/°C and an RI sensitivity of 2583.3 nm/RIU in the 1.333–1.343 range [29]. The structure is in fact a Mach-Zehnder interferometer and the intermediate section introduces a temperature dependent phase shift compensating the LPG temperature dependence.

## 3. Micro In-Line Mach–Zehnder Interferometers as Highly RI Sensitive Optical Fiber Platforms

### 3.1. In-Line All Fiber Mach-Zehnder Interferometers

As outlined in the introduction, the other highly RI sensitive and thus promising fiber structure for biosensing, including bacteria sensing, are all fiber-based Mach-Zehnder interferometers (MZIs). Comparing to LPGs where the sensor length is up to ca. 50 mm, the lengths of femtosecond laser fabricated micro-cavity MZIs can be as short as 10 µm [14,30]. Tapered MZIs have sensing lengths from 30 mm to 40 mm and are comparable to the lengths of LPGs. Their sensitivities to RI changes are of the order of 400 nm/RIU around RI of water [31], which is considerably less than that of DTP-LPGs. A MZI based on up-down tapers [32] offersa sensitivity of −310 dB/RIU, which again is comparable or less than a similar type of sensitivity of DTP-LPGs where it varies from 317 dB/RIU to 970 dB/RIU (Figure 3b and Figure 4b). A somewhat better sensitivity of 703 nm/RIU is reported for a multicore tapered sensor of 20 mm [33]. MZIs based on etched photonic crystal fiber sandwiched between standard SMF fibers exhibited a sensitivity of 360 nm/RIU for a 35 mm long section length and PCF fiber cladding reduced to 91 µm [34].

Compared to the above MZ structure the femtosecond laser drilled microcavity MZIs exhibit a small size of the order of tens of microns and RI sensitivity comparable to that of nanocoated LPGs. The microcavity can be of different shapes as schematically shown in the Figure 5: conical [14,35,36], trench [37,38], circular [39,40,41,42,43].

In all cases, the microcavity cuts partially through the core, so that part of the mode propagates in the core and another part in the cavity. The conical microcavity interferometer showed a RI sensitivity of about 9370 nm/RIU for a diameter of 51 µm [36], the trench cavity—about 10,800 nm/RIU around 1.333 (water) and −3243 nm/RIU around 1 (air) for trench lengths from 50 µm to 115 µm [37,44]. The circular micro-cavity interferometer of diameter 60 µm—about 12,000 nm/RIU around 1.333 [38,39,40]. These and similar structures are referred as microcavity in-fiber MZI–µIMZI.

### 3.2. Structure of Micro-interferometers in Transmission and in Reflection

The µIMZI is a hole carved across the cladding of a single-mode fiber penetrating partially into the core of RI *n*_0_ as shown in Figure 5a. The fundamental core mode propagating along the core has an approximately Gaussian distribution whose spread into the cladding is wavelength dependent, i.e., the higher the wavelength, the broader the distribution. When it reaches the cavity filled by a medium of lower RI (*n*), a part *p*(−1 Δ ≤ *p* ≤ 1) of the mode electrical field is coupled into the cavity propagating at a different speed. For *p* > 0 the power is coupled from the core to the cavity while for *p* < 0 it is coupled back to the core. Since *n* < *n*_0_ the beam is divergent, and losses occur [38]. At the other end of the microcavity a relative power *p* couples back to the core after having accumulated a phase difference with respect to the part propagating in the core whose relative part of the electric field is *q*.

Power conservation demands that *p*^2^ + *q*^2^ = 1. In case of losses *p*^2^ + *q*^2^ < 1.The coupling process in case of transmission is schematically represented as shown in Figure 6a, while that in reflection is shown in Figure 6b. The coupling coefficient *p* increases with wavelength since mode field broadens. The structures in transmission are referred to as µIMZI, while those in reflection as micro-cavity in-fiber reflective interferometer (µIRI).

#### Responses and Sensitivities of Micro-Interferometers

Ignoring losses due to scattering and diffraction, after accumulating a phase delay the light is coupled back to the core. If the input intensity is *I*_0_, the intensity at the output of the µIMZI in the case of no losses is defined as in Equation (3a) [36,37,38,39,40,44], while that of the µIRI is as in Equation (3b), and the accumulated phase difference is as in Equation (3c).
(3a)$$I_{T}=I_{1}+I_{2}+2\sqrt{I_{1}I_{2}}\cos\Phi \;\;\;\;I_{1}=p^{4}I_{0},\;I_{2}=q^{4}I_{0}$$
(3b)$$I_{R}={\left(I_{1}+I_{2}+2\sqrt{I_{1}I_{2}}\cos\Phi \right)}^{2}=I_{T}^{2}$$
(3c)$$\Phi =\frac{2\pi }{\lambda }\Delta n.l+\varphi \;\;\;\;\;\;\Delta n=n_{0}-n$$

In Equation (3a,b) *I*_1_ and *I*_2_ are the intensities of the field in the core and the cavity respectively, Δ*n* = *n*_0_ − *n* is the effective RI difference at wavelength *λ* between the core mode effective RI and that of the medium inside the cavity of effective length *l*_0_, *ϕ* being some constant phase shift accumulated because of microcavity shape imperfections.

A comparison between the responses of an interferometer in transmission and one in reflection shows that the losses in the reflective interferometer are twice higher in dB since from Equations (3a,b) log*I*_R_ = 2log*I*_T_. The advantage of a reflective microcavity interferometer is that it can be mounted, e.g., in a needle and be used as a tip sensor.

When a microcavity is drilled into the fiber, the exact positions of the minima cannot be fixed at will. However, for a practical device one would prefer to have the minimum at the right end of the spectral range to have sufficient spectral range for wavelength shifts caused by RI changes. To fine-tune the position of the minima extended microcavities can be used [38].

Both transmission and reflection structures show the same RI sensitivity [14,37] and free spectral range (FSR):(4a)$$S_{\Delta n}=\frac{d\lambda }{d\left[\Delta n(\lambda )\right]}=\frac{d\lambda }{d(\Delta n)}=\frac{\lambda }{\Delta n(\lambda )}=-\frac{\lambda }{n_{0}(\lambda )-n(\lambda )}$$
(4b)$$\Lambda =\frac{{\bar{\lambda }}^{2}}{\Delta n(\bar{\lambda })l_{0}}=S_{\Delta n}\frac{\bar{\lambda }}{l_{0}}\;\;\;\;,\;FSR=\frac{{\bar{\lambda }}^{2}}{\Delta n(\lambda )l_{0}}=S_{\Delta n}\frac{\bar{\lambda }}{l_{0}}\;\;\;\;\bar{\lambda }=\sqrt{\lambda_{m}\lambda_{m+1}}$$

The sensitivity *S_Δn_* (nm/RIU) is thus proportional to the wavelength of the traced spectral minimum and inversely proportional to the effective modal RI difference while the length *l*_0_ affects the free spectral range and not *S_Δn_*. A µIMZI and a µIRI of same length *l*_0_ will thus have the same sensitivity, but the minima of the latter will be twice deeper than those of the former.

### 3.3. Fabrication and Sensitivity Tuning of Micro in-Fiber Interferometers

#### 3.3.1. Fabrication Technologies of µIMZIs

Microcavity in-line Mach-Zehnder or reflective interferometers are usually fabricated using femtosecond lasers. The micromachining is similar to previously reported techniques [14,36,37,38,39,40,41,44] was performed using a Solstice Ti: Sapphire fs laser operating at *λ* = 795 nm as shown in Figure 7.

The cavities are usually of size *d* = 60 µm in diameter and *h*_0_ = 62 µm in depth fabricated in a SMF-28 fiber according to the procedure described in detail elsewhere [14,37,38,39,40]. In the experiments [39,40,41,42,43], the fiber was irradiated by 82 fs pulses at a repetition rate of 10 kHz. In order to make the microcavity, the laser beam is directed into a suitably designed micromachining setup based on the Newport µFab system. The system was equipped with a 20× lens, with NA = 0.30. Fiber transmission was monitored throughout the micromachining process with the NKT Photonics SuperK COMPACT supercontinuum white light source and Yokogawa AQ6370C optical spectrum analyzer in 1100–1700 nm spectral range. The fabrication process was controlled with an in-house developed software.

#### 3.3.2. Fine-Tuning of µIMZIs

Fine-tuning of the µIMZI includes positioning the minimum to be tracked at a desirable wavelength and increasing the sensitivity. The first approach is reported in [38] where it is proposed to machine an extension of the microcavity and by changing its width and depth a minimum can be tuned by 200 nm and the sensitivity increased by 32% reaching 26,880 nm/RIU in the 1.3333–1.3412 RIU. The second approach is to apply reactive ion etching in oxygen plasma [39]. Etching improves wettability and thus improves sensitivity from 11,377 nm/RIU to 11,800 nm/RIU around 1.333.

Thin Al_2_O_3_ film deposition gives the possibility [41] for a transition from volume to surface sensitivities, which gives the possibility for responses when surface RI changes due to the presence of bacteria, for example. The above-cited values for the sensitivities are measured in the 1400 nm to 1700 nm range and the insertion losses are around 10 dB. According to Equation (4a) sensitivity S*_Δ_*_n_ around 800 nm will be twice lower when compared to that at 1600 nm, which means that it would be of the order of 6000 nm/RIU. Since the fibers to be used must be single mode at 800 nm, the insertion losses would increase because of the smaller radius. However, tapering the fiber and drilling the microcavity in the tapered part decreases losses down to 5 dB and increases sensitivity up to 14,000 nm/RIU for SM800 fiber and up to 11,500 nm/RIU in SM600 nm [42].

One general advantage of microcavity interferometers is the extremely small amount of sensing volumes. For cavity diameters from *d* = 50 µm to 60 µm and a depth of *h* = 60 µm the volume is from *V* = 117 *pL* to *V* = 170 *pL*. Using nanocoating on the bottom for a surface sensitivity means that the outer cladding of the fiber can be reduced in a way that the microcavity depth reduces to about 10 µm which would reduce the sensing volume to less than 50 *pL*.

Table 1 summarizes the RI sensitivities of various optical fiber sensing platforms considered above. As seen, standard LPGs and interferometric structures cannot reach sensitivity above 2000 nm/RIU except for a sensor using evanescent field in an asymmetrical taper. LPGs around DTP can reach thousands of nm/RIU and additional etching and nano-coating can increase it to 20,000 nm/RIU. The µIMZIs on the other hand, usually exhibit sensitivity from 9000 to 11,000 nm/RIU without additional processing or coating. Chemical etching and cavity extension can increase sensitivities to record values of 17,200 and 27,000 nm/RIU.

## 4. Bacteria Detection Using LPGs

In literature, throughout the years we can find several LPG-based bacteria detection systems. They differ in the design of the sensing part, the choice of the bioreceptor, and surface functionalization. However, each fiber optic biosensor essentially comprises a biorecognition element—a bioreceptor which drives the selectivity and specificity of the sensor, chemical functionalization to immobilize the bioreceptor, and properly orient it on the sensor’s surface, and a transducer able to convert the target-receptor binding into a measurable and detectable signal (Figure 8).

In our work, we will divide the sensors based on the target—whether the described method aim detection of whole-cell (direct) or just a part of bacteria (indirect)—and in terms of the biorecognition element.

### 4.1. Direct Methods

Direct methods refer to the detection of whole bacteria cells. When early and fast diagnosis is required, this system is one of the best ways to identify the pathogenic strain of bacteria. The main advantages of this approach are toleration of the whole-cells to the environmental changes such as temperature, pH, etc. in contrast to isolated structures; the ease of the isolation of the bacteria cells from natural sources; and no need for extensive preparation of the samples before the measurement process. Nonetheless, the development of the sensing system for the whole bacteria cell is challenging. We need to take into consideration the size of the bacteria, which is much larger than other analytes, such as proteins, DNA aptamers, and its surface structures, e.g., pili and flagella, which can disturb measurements. Moreover, a number of the outer structures on the bacteria surface can serve as biorecognition/binding places and may lead to nonspecific interactions with the sensor’s surface. Moreover, depending on the form, which we target to detect, whether it will be live bacteria, dry mass, or heat-killed organism, they will act differently during the assay. Thus, to avoid the false-positive interactions the biorecognition element on the sensor’s surface supposed to be well chosen, as it will be responsible for specificity and partially for the efficiency of the whole experiment.

#### 4.1.1. Bacteriophages

The first published attempt to bacteria detection using LPG was done utilizing the T4 bacteriophages as a biorecognition element. As it is a natural and very specific enemy of the *E. coli* bacteria, it could be successfully introduced to the whole-cell bacteria detection. In [43], Smietana et al. reported the LPG with the sensitivity ~600 nm/RIU and physically adsorbed bacteriophages on the LPG surface. Due to the relatively low sensitivity of the platform, proof-of-concept character of the studies, and the non-covalent link between the surface and the bioreceptor, only high concentrations of *E. coli* were detected, with no connection between the targeted bacteria concentration and obtained spectral shift.

One year later, Tripathi et al. [45] reported the same approach, however, to assure the covalent binding of the T4 phages the chemical functionalization of the surface was introduced. The silanization with APTES and activation of the obtained −NH_2_ groups using glutaraldehyde provided −COOH groups, which could be easily bonded to the −NH_2_ groups present on the T4. As the sensor had higher sensitivity reaching ~2000 nm/RIU, and the bioreceptor was covalently bound to the sensor’s surface, the sensing schema was tested against different concentrations of *E. coli* bacteria in colony-forming units (CFU) and resulted in 10^3^ CFU/mL limit of the detection.

Although the T4 bacteriophages are very specific, they also have limitations. Considering highly complex structure, they are prone to mechanical damages and, what is more, even after chemical functionalization of the LPG’s surface we cannot be sure about their orientation on the sensor’s surface. In addition, the T4 is the lytic phage, which means that after binding the bacteria cell and infection, the *E. coli* cells are destroyed releasing newly replicated phages, which, in turn, can lead to false-negative/positive signals.

In response to that, Chiniforooshan et al. [46] reported a comparison of the *E. coli* detection using T4 and MS2 phages. The difference in shape, robustness, place of the attachment, the infection cycle of the MS2 and surface functionalization performed utilizing gas instead of liquid phase resulted in lowering the limit of the detection from 10^3^ to 100 CFU and higher stability of the measurements. The sensitivity of the LPG utilized during this experiment reached ~7000 nm/RIU in the RI range from 1.33–1.34 RIU.

#### 4.1.2. Adhesins—Receptor Binding Proteins

The relatively large size of bacteriophages and drawbacks associated with them can be a problem for some biosensing applications. Therefore, in some cases, just receptor binding proteins located on the phages’ fiber tails can be applied. These proteins are responsible for specific recognition of host-bacteria and one of their biggest advantages is the possibility of introducing changes to their sequence/structure [47]. Without disturbing and altering their binding affinity, we can improve protein’s orientation on the sensor’s surface [48]. The adhesin utilized on LPG for label-free *E. coli* bacteria detection is protein gp37 obtained from phage T4.

Brzozowska et al. [49] and Koba et al. [50] reported adhesin’s gp37 ability to detect whole bacteria cells and presented its specificity. In both cases, for the experiment LPGs with RI sensitivity reaching ~3000 nm/RIU have been utilized. The functionalization process applied in these experiments was intended for proteins containing histidine (His) tag. The chemical functionalization of LPG included silanization of the surface in GPTMS solution. Next, the LPG was immersed in 20 mM (milimolar) iminodiacetic acid in NaHCO_3_ buffer. The last step included the immobilization of the nickel ions on the sensor’s surface. Such prepared surface was ready to attach the bioreceptor—adhesin gp37. The goal of these studies was to determine the sensor’s specificity; thus, it was verified with many bacterial strains such as *S. enterica*, *K. pneumoniae*, *E. coli* K12, or *Lactobacillus* 919. All the strains were detected in one concentration—namely 100 µg/mL as the lowest limit of detection (LOD) was not the priority. The research has clearly proven the possibility of using adhesins in place of bacteriophages for specific bacteria strain recognition.

Next work utilizing the adhesin gp37 as a bioreceptor was reported by Piestrzynska et al. [51]. Here, a label-free biosensing application of LPGs optimized in RI sensitivity by deposition of thin tantalum oxide (TaO_x_) overlays was discussed. The TaO_x_nano-coated LPGs utilized during the experiment showed extremely high RI sensitivity ~11,500 nm/RIU in RI range 1.335–1.345 RIU. Besides enhancing the RI sensitivity, TaO_x_ was also expected to provide the chemical and mechanical stability during the surface functionalization or biosensing procedure. For the *E. coli* bacteria sensing experiment, the TaO_x_ nano-coated LPG was first subjected to surface silanization with triethoxysilylpropyl succinic anhydride (TESPSA). Next, the formation of an amide bond between amine groups in the Ni-NTA complex and groups of succinic anhydrides on the surface of the fiber enabled attachment of the adhesin containing His-tag at the N-terminus. Despite highly increased initial sensitivity of the nanocoated LPGs, the LOD reached just 10^3^ CFU/mL. The experiment proved that during the work with such sensitive sensors, the functionalization process, along with the size of the bioreceptor and the biological target is crucial. Thus, every modification of such LPG needs to be taken into consideration because it may significantly reduce initially optimized RI sensitivity.

#### 4.1.3. Antibodies

The next group of the LPG-based sensors includes devices utilizing antibodies (Abs) as bioreceptors. Abs are also known as immunoglobulins (Ig), are large, “Y-shape” proteins with dimensions approximately 14 × 10 × 4 nm [52]. Each Ab molecule has its unique structure that enables it to bind specifically to its corresponding target. They have several advantages such as high specificity, versatility and thanks to the ease of integration into different systems, Abs were combined with numerous sensor formats. Once they are selected and secreted, the immobilization process becomes crucial for their efficiency. The control over the position and orientation of Abs during the covalent binding is challenging. In an ideal scenario, antibodies should be immobilized in their native form and oriented to maximize complementary binding, without the need for introducing functional groups or special tags/labeling. To date, no method genuinely provides the ideal scenario [53]. Moreover, depending on the target, Abs can provide different immobilization efficiency of the detected pathogen [54] and they are prone to physical, chemical, and enzymatic damages. However, despite mentioned drawbacks, so-called immunosensors have been regarded as a gold standard in diagnostics and environmental monitoring.

The use of Abs was selected for the detection of bacteria like *E. coli* and *S aureus*. In the detection of *S. aureus*, Bandara et al. [55] functionalized the LPG surface using layer-by-layer (LbL) deposition of self-assembled polyelectrolyte multilayers. Immersion of the fiber in PCBS resulted in terminal carboxyl groups, which were then covalently conjugated to the Abs. Chosen Abs were specific to penicillin-binding-protein 2a of methicillin-resistant (MR) staphylococci. The binding of the target bacteria to the sensor was detected as a change in the transmitted power through the fiber at the wavelength of peak attenuation of the turning around point-LPG (1550 nm). In a buffer solution, a LOD of 100 CFU was reached, but they also showed the selectivity of the sensor testing 36 different strains of MR staphylococci (all at 10^4^CFU concentration).

The same approach in terms of LPG’s functionalization was also implemented by Fang et al. [56] in the microfluidic-LPG schema. However, in this work polyelectrolyte, functional coatings PAH/PAA were additionally modified to selectively facilitate the bacterial adhesion on the surfaces. Moreover, the functional coatings not only increase the initial rate of bacterial adhesion onto the surfaces (as the experiment was conducted in the constant flow), but also improve the sensitivity of LPG and thus the detection limit for *S. aureus* (calculated 224 CFU/mL, experimental range 10^4^–10^8^ CFU).

Another configuration of LPG used in immunosensing presented by Kaushik et al. [57] is a combination of two identical chirped LPGs for *E. coli* detection. The described sensor with an inter-grating space works like a Mach-Zehnder interferometer. To immobilize monoclonal Abs the surface was silanized with APTES to generate the −NH_2_ groups and were activated with EDC/NHS. This results in the activation of carboxyl groups of anti-*E. coli* antibodies to form covalent bonding with the amine groups on the sensing probe via carbodiimide cross linker chemistry. Such prepared immunosensor was then exposed to different concentrations of *E. coli* (10 CFU/Ml–60 CFU/mL) and the obtained spectral shift for the lowest concentration reached 0.2 nm for 10 CFU/mL (calculated LOD 7 CFU).

#### 4.1.4. Carbohydrates

Another and a relatively new approach to bacteria detection in a view of bioreceptor is the use of carbohydrates. This system is based on the natural affinity of bacteria to specific carbohydrates, a moiety occurring in almost every living organism, which is a basis of the bacteria-host cell recognition. Therefore, they are able to naturally attract bacteria, facilitate biofilm formation, and promote bacteria adhesion [24]. In this sensing schema, the detection of pathogenic bacteria is carried out via protein-carbohydrate interactions, which are very similar to the antibody-antigen, or enzyme-substrate reactions.

Celebanska et al. [58] reported the LPFG-based system for *E. coli* detection. During the experiment, highly sensitive ~7000 nm/RIU LPG was used. The fiber optic platform was functionalized with APTES by a gas phase method. Next, the amine groups were activated by a homobifunctional cross linker PDITC to covalently bind the chosen carbohydrate—*α*-mannose (APMAN)—on the sensor’s surface. The sensor was examined with specific *E. coli* bacteria within the range 10^3^ to 10^8^ CFU/mL. Despite the very high sensitivity of the LPFG platform, the experimental LOD reached just 10^3^ CFU. Moreover, the much broader specificity in comparison to the bacteriophages or antibodies disqualifies the carbohydrates from one strain-targeted tests; however, such combination could be used as an entry test for potentially contaminated samples.

### 4.2. Indirect Methods

Indirect methods refer to the detection of chosen parts of bacteria. Depending on the type of bacteria, we can target different surface structures such as lipopolysaccharides (LPSs), outer membrane proteins (OMPs), porins, or glycoproteins. Even though they need isolation and preparation before the use, they are a very good starting point for initial tests, which are in most cases the base for further phases targeting the detection of the whole microorganisms. What is more, it is a very good solution for laboratories without special facilities needed to handle the work with whole bacteria (especially pathogenic strains). To date, three groups reported the indirect detection of *E. coli* using an LPG-based system.

#### 4.2.1. Adhesins—Receptor Binding Proteins

The first work describes the use of the phage T4 adhesin gp37 mentioned in Section 4.1.2. Brzozowska et al. [58] showed the detection of *E. coli* LPS using a highly sensitive platform (3000 nm/RIU). The adhesin was again covalently bound to the surface by histidine-tag using nickel ions and was tested in LPS concentration equal to 0.2 mg/mL. Koba et al. [59] utilized the same approach to show the stability and reusability of the presented sensing system. Both presented works were just an introduction to use the adhesin as a bioreceptor for whole-cell bacteria detection, which was described in later articles.

#### 4.2.2. DNA Aptamers

Indirect detection was also described in the configuration where DNA aptamers were employed as a bioreceptor. An aptamer is a short (25 to 90 bases [60]), single-stranded nucleic acid sequence DNA or RNA, which can attach to a variety of targets like bacteria, viruses, proteins, drugs, or even heavy metal ions. Thanks to their three-dimensional structure, the aptamers can bind the target with high efficacy and specificity—comparable to, or even higher than Abs, what may be caused by aptamer’s capability of folding upon binding target [61]. They provide exceptional flexibility and convenience regarding the engineering of their structure what have led to the development of DNA aptamers with enhanced affinity, specificity, and stability [62]. In comparison to Abs, aptamers are structurally stable across a wide range of temperatures and storage conditions. Due to their outstanding binding characteristics, stability and possibility for easy modifications, aptamers have all the necessary qualities to be used as the basis of detection elements. Surprisingly, despite the availability of aptamers recognizing a wide range of pathogens, a relatively low number of them have yet been used in biosensing.

Queirós et al. [63] presented an LPG-based aptasensor for selective *E. coli* outer membrane protein detection. To covalently bind the DNA, the LPGs surface was functionalized using two different processes: an electrostatic method, using PLL as cationic polymer and a covalent method, using APTES. The LPG with the sensitivity of ~150 nm/RIU was tested with known protein concentration equal to 0.1 to 10 nm. In the end, both functionalization methods gave an identical performance in terms of detection; however, polycationic functionalization presented better results in terms of reproducibility and uniformity of functionalization.

## 5. Bacteria Detection Using µIMZI

LPGs were tested throughout the years also for the detection of many different bio-applications. Although they appeared effective and reached very low limits of detection, many challenges remain valid and urgent for the creation of successful biosensing systems. Common limiting factors of the LPGs’ are their capability for multiplexing or reproducibility, especially more complex platforms. Moreover, this type of sensor having 4–5 cm length of the fiber usually requires at least 500 µL of the examined sample, which can be problematic when only small volumes of analyte are available. Most of the LPG sensors are also sensitive towards temperature, which is an essential disadvantage since the temperature is one of the most disrupting factors in RI measurements. Despite generally good performance, there are still drawbacks and open questions about the most precise and reliable detection. Identification of small molecules in *pl* volumes—possibly up to single-molecule—undisturbed by external factors, along with miniaturization of the device is still a challenge. Therefore, constant attention is paid to developing new, cutting-edge devices that will be capable of solving these challenges.

One of the relatively new classes of sensing devices are structures based on in-fiber microcavities [15]. The presence of tens of micrometers diameter cavity micromachined in the fiber is the defining feature of these platforms. Although the microcavity-based sensors are known for almost a decade and have been reported by few groups in different shapes and for different applications, the subject of using it for specific biosensing has been reported just once so far. The only work using this sensing scheme for specific bacteria detection up to date was published by Janik et al. [64]. Microcavity in-line Mach-Zehnder interferometer with 60 um diameter and sensitivity reaching 15,000 nm/RIU was functionalized with APTES from a gas phase assuring uniform chemical layer on the sensor’s surface. Activated by GLU −NH_2_ groups were able to covalently attach chosen bioreceptor—MS2 bacteriophages. As described in Section 3, due to their properties, they evenly cover the surface of the microcavity. Such a prepared sensor was tested with different *E. coli* bacteria concentrations ranging from 100 to 10^8^ CFU and was able to detect 100 CFU of live bacteria in liquid volume as low as picoliters. Considering novel and miniature design, superior sensitivity, unique feature to measure volumes as low as 117–170 picoliters as shown above, temperature insensitivity, an in-fiber microcavity structure, and a fiber tip design would be a good alternative with a very high potential in biosensing.

Table 2 below summarizes the reported *S. aureus* and *E. coli* bacteria sensing results with LPGs and µIMZI.

## 6. Conclusions

In this paper, we presented the principle of operation of two highly RI sensitive platforms, i.e., long-period gratings operating around dispersion turning point (DTP-LPGs) and microcavity in-line Mach-Zehnder interferometer (µIMZI). Different types of Mach-Zehnder interferometers and of LPGs have been reviewed and their sensitivities have been compared. Comparison shows that in terms of sensitivity to RI the most promising structures are DTP-LPGs and µIMZIs with typical sensitivities around water RI from 2600nm/RIU to 11,000 nm/RIU and around 10,000 nm/RIU for µIMZI. With additional fine-tuning procedures, these sensitivities can be increased to correspondingly 20,000 nm/RIU and 27,000 nm/RIU. However, while sensitivity of DTP-LPGs reduces thereafter, that of µIMZI increases. Moreover, so far, DTP-LPGs have been demonstrated in a transmission mode, while microcavity MZIs have been demonstrated in reflection mode with a sensitivity of 12,000 nm/RIU.

Up to now, the DTP-LPGs have been more intensively explored as fiber-based sensing platforms for direct and indirect *E. coli* and *S. aureus* bacteria detection. A variety of surface functionalizations with different receptors as bacteriophages, adhesins, antibodies, and carbohydrates, and DNA aptamers, have been tested. Ranges over four orders of magnitude (10^2^–10^6^ CFU/mL and 10^4^–10^8^ CFU/mL) for *S. aureus* detection and over six orders of magnitude (10^3^–10^9^ CFU/mL) for *E. coli* have been demonstrated. Only one report so far has demonstrated *E. coli* detection of six orders of magnitudes (10^2^–10^6^ CFU/mL) using µIMZI. Taking into account the higher sensitivity of µMZIs and the possibility to fabricate them in reflection mode at the tip of a fiber, makes them promising for highly sensitive sub-nano- and picoliter dip biosensors mounted in medical needles.

## Figures and Tables

**Figure 1 sensors-20-03772-f001:**
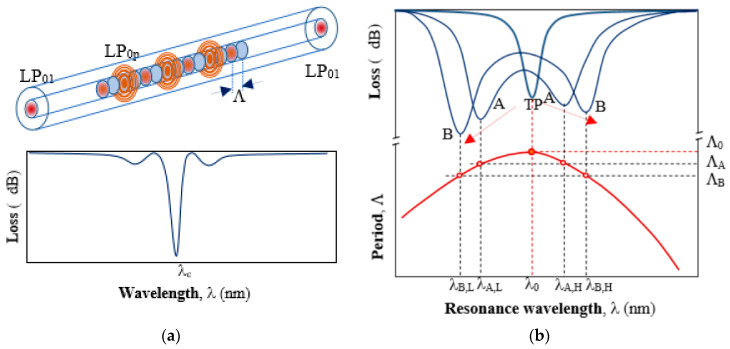
A long-period gratings (LPG)and its responses: (**a**) coupling between the fundamental LP_01_ core mode and a higher order LP_0p_ cladding mode along the grating and typical transmission spectrum, away from dispersion turning point (DTP) with a minimum at resonance wavelength *λ*_c_. (**b**) The relation between the period *Λ* and the resonance wavelength *λ*_0_ of an LPG around the DTP at a period *Λ*_0_ (lower curve). Changing the period *Λ* of the grating to *Λ*_A_ and *Λ*_B_, the spectrum of the LPGs splits and shifts in opposite directions as shown in (**b**) above.

**Figure 2 sensors-20-03772-f002:**
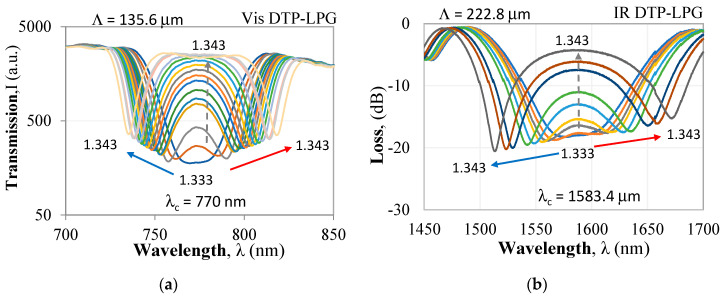
Spectral response of DTP-LPGs to refractive (RI) changes: (**a**) SM600-based LPG in the visible/near infrared (Vis DTP-LPG); (**b**) SMF-28-based LPG in the infrared (IR) DTP-LPG.

**Figure 3 sensors-20-03772-f003:**
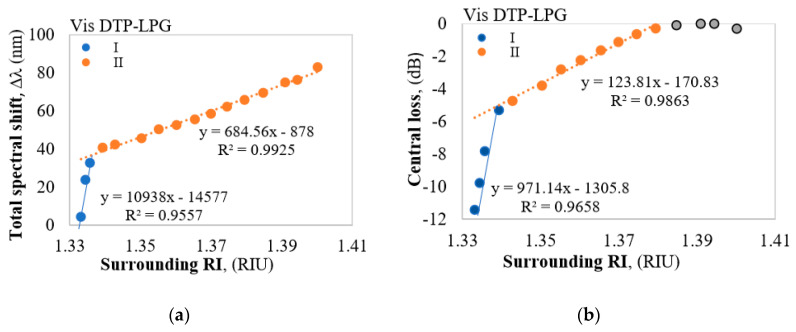
Responses of the Vis DTP-LPG: (**a**) Δ*λ* vs. RI; (**b**) Δ*I_DTP_* vs. RI.

**Figure 4 sensors-20-03772-f004:**
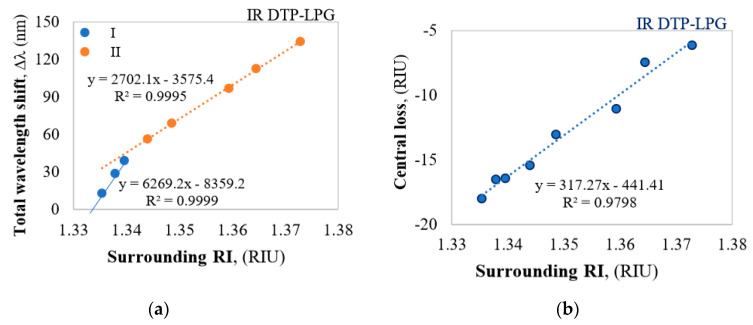
Responses of the IR DTP-LPG: (**a**) Δ*λ* vs. RI; (**b**) Δ*I_DTP_* vs. RI.

**Figure 5 sensors-20-03772-f005:**
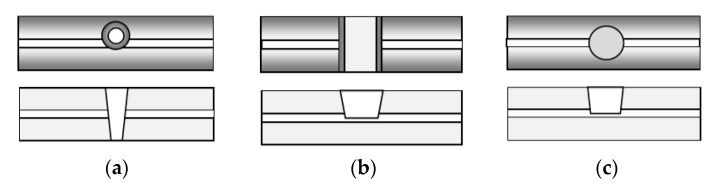
Top view and cross-section of the three basic shapes of microcavities: (**a**) conical; (**b**) trench; (**c**) circular.

**Figure 6 sensors-20-03772-f006:**
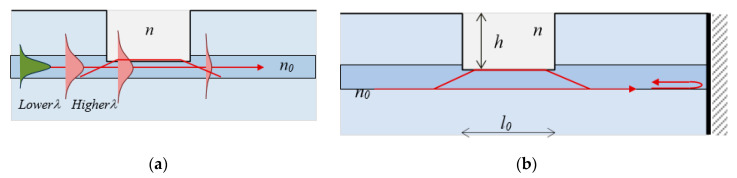
Microcavity interferometers: (**a**) core mode redistributionin a fiber with a microcavity; (**b**) mode coupling in a reflection scheme.

**Figure 7 sensors-20-03772-f007:**
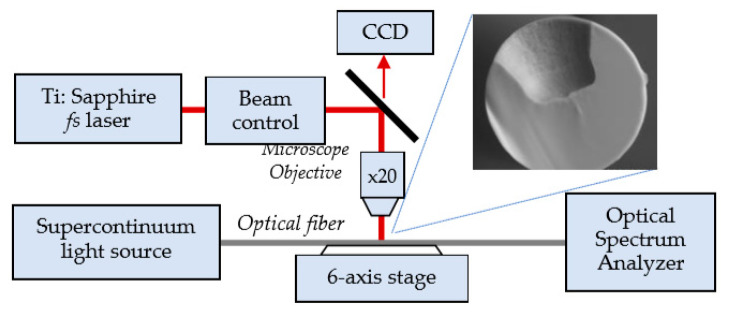
Schematic representation of the experimental set-up for the fabrication of a microcavity in a fiber using a femtosecond laser. A SEM view of the cross section of the microcavity.

**Figure 8 sensors-20-03772-f008:**
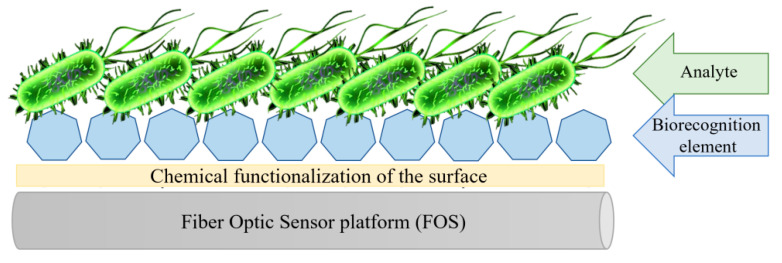
Schematic representation of the fiber optic biosensor.

**Table 1 sensors-20-03772-t001:** Comparison of Mach-Zehnder interferometers, LPGs, and in-line Mach-Zehnder interferometers (µIMZI), and their sensitivities around water RI of 1.333.

Structure	Specific Features	*S*_Δ*n*_ [nm/RIU]	Reference
LPG	Double resonance	750	[10]
LPG	SiO_2_ nanoparticle mesoporous thin films	1927	[9]
Twin-core	SMF-Twin core-SMF	1973	[17]
D-shaped	Nanocoated D-shaped fiber	1566	[19]
SMS	*fs* induced spiral MSF	2144	[21]
MZI	PCF	100–150	[16]
MZI	Etched PCF	360	[34]
Tapered	SMF	415	[31]
Tapered	Multicore fiber	703	[33]
Taped fiber	Asymmetrical taper evanescent field	3390–3914	[18]
DTP-LPG	Cladding etching and nano-coating	20,000	[26]
µIMZI	Conical	9370	[35]
µIMZI	Conical	10,537	[14]
µIMZI	Conical and chemical etching	17,197	[36]
µIMZI	Plasma-processed	11,800	[39]
µIMZI	Trench	10,000	[44]
µIMZI	Trench	10,780	[37]
µIMZI	Circular, thin film deposition	12,390	[41]
µIMZI	Circular, extended	26,882	[38]
µIRI	In reflection	13,865	[40]
µIMZI	Vis/NIR tapered	12,000	[42]

**Table 2 sensors-20-03772-t002:** Fiber optic biosensors for bacteria detection.

Sensor Type	Sensitivity of the Sensor [nm/RIU]	Type of Detected Bacteria	Range of the Detection [CFU/mL]	Surface Functionalization	Receptor	Ref
DIRECT METHODS						
**Chirped-LPG**	-	*E. coli*	10–60 CFU/mL	APTES+EDC	Ab	[65]
**LPG**	7200	*E. coli*	10^3^ CFU/mL	APTES+PDITC	APMAN	[58]
**2 LPG**	1929	*E. coli*	10^2^–10^5^ CFU/mL	APTES + GLU	T4 phages	[66]
**LPG**	570	*E. coli*	10^8^ CFU/mL	-	T4 phages	[45]
**LPG**	-	*E. coli*	100 CFU/mL	APTES+GLU	MS2 phages	[47]
**LPG**	-	*S. aureus*	10^4^–10^8^ CFU/mL	ISAM ([PAH/PAA]_10_PAH)	Ab	[56]
**LPG**	-	*S. aureus*	10^2^–10^6^ CFU/mL	ISAM (PCBS)	Ab	[55]
**LPG**	2600	*E. coli*	100 µg/mL	GPTMS+Ni + His-Tag	adhesin gp37	[50]
**LPG**	3000	*E. coli*	100 µg/mL	GPTMS +Ni + His-Tag	adhesin gp37	[51]
**LPG**	2321	*E. coli*	10^3^–10^9^ CFU/mL	APTES+GLU	T4 phages	[46]
**LPG**	11,000	*E. coli*	10^3^–10^9^ CFU/mL	TESPSA+Ni + His-Tag	gp37	[52]
**µIMZI**	15,000	*E. coli*	10^2^–10^8^ CFU/mL	APTES+GLU	MS2 phages	[63]
INDIRECT METHODS						
**LPG**	6947.4	LPS *E. coli* B	0.2 mg/mL	GPTMS+Ni + His-Tag	adhesin gp37	[59]
**LPG**	2600	LPS *E. coli* B	0.2 mg/mL	GPTMS+Ni + His-Tag	adhesin gp37	[67]
**LPG**	150	OMPs *E. coli*	0.1–10 nM	ISAM (PLL) APTES	DNA aptamer	[63]

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
