# Peer review of "Long-Period Gratings and Microcavity In-Line Mach Zehnder Interferometers as Highly Sensitive Optical Fiber Platforms for Bacteria Sensing"

_sensors, 2020, doi:10.3390/s20133772_

Round 1
Reviewer 1 Report
This seems to be a review paper, rather than a paper with new results. It depends on the policy of the journal whether this is acceptable.
I also want to mention that I have some background in optical sensors, but no background in the chemical or biological aspects. Therefore, some parts of the paper are hard to judge for me. However, the optical part of the paper is written quite well and clear.
This being said, it is reported that the main sensors discussed in the paper also suffer from temperature dependence, which is typical for refractive index based sensing. The authors would do good to address in their review paper also how one can compensate for the temperature dependence instead of just mentioning it.
Finally the language or spelling could be improved here and there. There seems to be also some problem with the numbering of Figures 6 and 7 or references to these figures. Sometimes the text refers to Figure 7 where it should be Figure 6.
Author Response
Response to reviewer 1
Dear Reviewer,
Thank you very much for your comments and recommendations. Our responses to your comments in red.
Comments and Suggestions for Authors
This seems to be a review paper, rather than a paper with new results. It depends on the policy of the journal whether this is acceptable.
I also want to mention that I have some background in optical sensors, but no background in the chemical or biological aspects. Therefore, some parts of the paper are hard to judge for me. However, the optical part of the paper is written quite well and clear.
This being said, it is reported that the main sensors discussed in the paper also suffer from temperature dependence, which is typical for refractive index based sensing. The authors would do good to address in their review paper also how one can compensate for the temperature dependence instead of just mentioning it.
A new subsection 2.3.3 has been introduced to address the issue with temperature stabilization and compensation. We have also added a new reference describing a wide range temperature insensitive DR LPG arrangement, namely:
- M. Tripathi, J. Bock, A. Kumar, W. and P. Mikulic, Temperature insensitive high-precision refractive-index sensor using two concatenated dual-resonance long-period gratings, Opt. Lett.2013, 38, 1666-1668
Finally the language or spelling could be improved here and there. There seems to be also some problem with the numbering of Figures 6 and 7 or references to these figures. Sometimes the text refers to Figure 7 where it should be Figure 6.
We have also corrected the numbering of figures and text accordingly.

Reviewer 2 Report
The manuscript entitled “ Double resonance LPGs and microcavity in-line Mach Zehnder interferometers as highly sensitive platforms for bacteria sensing – a review” should show the state of the art of two fiber platforms, long period gratings around turning point and micro-cavity In-line Mach-Zehnder Interferometers applied to bacteria sensing. However, there are a number of issues that in my opinion should be improved before publication:
- Starting from the title itself, in my opinion it is somehow confused and is not attractive at first reading. It would be better to change it, is only a personal comment.
- This review is mainly for E. Coli bacteria. There are some references to the S. aureus but if the authors want to perform a good review about sensors to detect bacteria it would be nice to read about a wider number of bacteria type detection and not to some of them.
- Figure 2 – The legend (a) and (b) is missing
- The section 2.3 is very weak in my opinion. Should be improved.
- Line 355 – The published paper that the authors are explaining should be referenced here.
- The section 6, in line 511, 6. Future alternatives – bacteria detection using μIMZI, what was the purpose? If this is intended to be a review of micro IMZIs, they can’t be future alternatives.
- The reference 17 is not right.
Overall, this manuscript is pointing to long period fiber gratings and its detection to E- coli bacteria. If the authors wants to keep the claim to a review of two different types of sensing configurations, the long period fiber gratings and the interferometers, applied to the detection of bacteria, they must perform an exhaustive state of the art of these two configurations applied to the detection of several types of bacteria. Which, in my opinion, was not accomplished yet.
Besides, the structure of the manuscript is not well organized, in my opinion.
Author Response
Dear Reviewer,
Thank you very much for your valuable remarks and comments.
Below in red are our responses.
Comments and Suggestions for Authors
The manuscript entitled “ Double resonance LPGs and microcavity in-line Mach Zehnder interferometers as highly sensitive platforms for bacteria sensing – a review” should show the state of the art of two fiber platforms, long period gratings around turning point and micro-cavity In-line Mach-Zehnder Interferometers applied to bacteria sensing. However, there are a number of issues that in my opinion should be improved be for publication:
Starting from the title itself, in my opinion it is somehow confused and is not attractive at first reading. It would be better to change it, is only a personal comment.
The title focuses on two of the most sensitive fiber platforms for a particular biosensing application. We have changed it to: “Long-period gratings and microcavity in-line Mach Zehnder interferometers as highly sensitive optical fiber platforms for bacteria sensing” and we believe now it will be clearer.
This review is mainly for E. Coli bacteria. There are some references to the S. Aureus but if the authors want to perform a good review about sensors to detect bacteria it would be nice to read about a wider number of bacteria type detection and not to some of them.
While writing the paper we did an extensive search on sensing other types of bacteria with these platforms and our list of references revealed all. Following your remark we did another review of literature and the results were the same. Thus, to our best knowledge, the cited papers are the only ones on bacteria detection using LPGs and micro in-line Mach-Zehnder Interferometers.
Figure 2 – The legend (a) and (b) is missing
We have added arrows showing the direction spectral of changes in this figure. These correspond to the notation in Fig. 1.
The section 2.3 is very weak in my opinion. Should be improved.
We accept the critical remark. We have extended sub-section 2.3.1 and have compared the advantages and disadvantages of different types of LPG writing technologies.
We have also added another subsection 2.3.3 on temperature stability and compensation.
Line 355 – The published paper that the authors are explaining should be referenced here.
All commented papers are cited. New numbers from [45] to [47].
The section 6, in line 511, 6. Future alternatives – bacteria detection using μIMZI, what was the purpose? If this is intended to be a review of micro IMZIs, they can’t be future alternatives.
We agree μIMZIs cannot be qualified as alternatives. It is just another approach which offers a sub-nL capability and is attractive because of its much higher sensitivity. We have removed this part of the section title.
The reference 17 is not right.
Reference corrected.
Overall, this manuscript is pointing to long period fiber gratings and its detection to E- coli bacteria. If the authors wants to keep the claim to a review of two different types of sensing configurations, the long period fiber gratings and the interferometers, applied to the detection of bacteria, they must perform an exhaustive state of the art of these two configurations applied to the detection of several types of bacteria. Which, in my opinion, was not accomplished yet.
We have reviewed the basic SRI sensitive fiber platforms used for biosensing applications with their sensitivities compared in Table 1. The Table reveals that the most sensitive platforms are DR LPGs and the μIMZIs. We therefore concentrate on these platforms for bacteria sensing applications. As commented above, we did not find other references on the application of these structures for bacteria sensing. We believe we can attract the attention for other bacteria detection using these platforms
Besides, the structure of the manuscript is not well organized, in my opinion.
The structure is based on the logic of first presenting the fiber sensing platforms and then the applications for bacteria sensing. In this way the platforms can be compared by sensitivity, concentrate on two of them and then their applications which in turn, are compared.

Reviewer 3 Report
References are needed in the first paragraph of introduction.
Line 69: add a space between called and turn.
Line 72: spelling check the first word
Line 78: add a space between sensor and are
Line 84: add 'range' after 1.3373-1.4345
Restructure the sentence in lines 94 to 97
Line 99, add a coma between LPG and only
Add legend to figure 2 for different lines
DR-LPG in figure 2 is not defined.
Indicate the periods and center wavelengths of the two LPGs in figure 2.
Typo in figure 2 caption: (IR DRLPG) not (IT DRLPG)
Restructure the sentence on lines 155 - 157 and refer it to figure 3.
Restructure the sentence on lines 162 - 164.
Line 172: "With the exception of the UV laser method using amplitude 172 masks, the rest are point-by-point writing methods" define 'the rest'
Provide more details on fabrication technologies of LPGs in lines 170-174
Properly placed coma is needed in lines 182 - 184.
line 195: wrong reference to fig.4a and fig.5b
Line 213: include the diameter of the circular type
Line 217: there is no figure 7a
In figure 6, there is no c)
equation 3c has l while line 248 uses l0.
Line 251: reference to the equations 4a and 5 is wrong
Check the sentence in line 258
Line 272: eliminate the word usually and provide a range of the diameter and depth
Figure 8 should be referenced in the text
Line 298: a proper period should be added.
Line 303: What is 10 in 'According to (10) sensitivity'?
Line 310 says the typical diameter is 50 micron, while line 272 says the diameter is usually 60 micron. What is the diameter?
Line 337: should be 4.1
Line 370 and 372, eliminate 'what is more'.
Line 421: why use past tense?
Do not use antibodies and Abs whenever you feel like. Be consistent
line 534, there is no section A
Check the section numbers
Author Response
Response to Reviewer 3
Dear Reviewer,
Thank you very much for your detailed comments, remarks and suggestions.
Our responses are given in red after your remarks.
References are needed in the first paragraph of introduction. Three references are added.
Line 69: add a space between called and turn. Space added.
Line 72: spelling check the first word. Corrected
Line 78: add a space between sensor and are. Space added
Line 84: add 'range' after 1.3373-1.4345 “Range” added
Restructure the sentence in lines 94 to 97 Split in two sentences.
Line 99, add a coma between LPG and only Coma added
Add legend to figure 2 for different lines Arrows similar to those in Fig. 1b) showing the direction of shifts have been added
DR-LPG in figure 2 is not defined. DR-LPG is introduced on lines 124, 141 and 145
Indicate the periods and center wavelengths of the two LPGs in figure 2. Periods and center wavelengths indicated in the figure and in text
Typo in figure 2 caption: (IR DRLPG) not (IT DRLPG) Typo corrected.
Restructure the sentence on lines 155 - 157 and refer it to figure 3. Sentence restructured.
Restructure the sentence on lines 162 - 164. Sentence restructured, now 163-64.
Line 172: "With the exception of the UV laser method using amplitude 172 masks, the rest are point-by-point writing methods" define 'the rest The text has been reworked to avoid confusion.
Provide more details on fabrication technologies of LPGs in lines 170-174. Details provided.
Properly placed coma is needed in lines 182 - 184. Coma placed.
line 195: wrong reference to fig.4a and fig.5b Corrected. Now Line 211
Line 213: include the diameter of the circular type. Diameter included (now line 229)
Line 217: there is no figure 7a Error corrected (now line 219)
In figure 6, there is no c) Corrected
Equation 3c has l while line 248 uses l0. Corrected
Line 251: reference to the equations 4a and 5 is wrong Corrected to (3a) and (3b)
Check the sentence in line 258 Corrected SΔn
Line 272: eliminate the word usually and provide a range of the diameter and depth Done
Figure 8 should be referenced in the text Done
Line 298: a proper period should be added. Done
Line 303: What is 10 in 'According to (10) sensitivity'? Corrected to (4a)
Line 310 says the typical diameter is 50 micron, while line 272 says the diameter is usually 60 micron. What is the diameter? The evaluation for a cavity from d = 50 μm to d = 60 μm is done. In our experiments it was usually 60 μm, in references 50 μm can be encountered
Line 337: should be 4.1 Done
Line 370 and 372, eliminate 'what is more'. Corrected to “In addition”,
Line 421: why use past tense? Corrected to present tense
Do not use antibodies and Abs whenever you feel like. Be consistent Abs used
Lline 534, there is no section A Done Section 3
Check the section numbers Corrected.

Round 2
Reviewer 2 Report
The manuscript initially entitled “ Double resonance LPGs and microcavity in-line Mach Zehnder interferometers as highly sensitive platforms for bacteria sensing – a review” was changed to “Long-period gratings and microcavity in-line Mach Zehnder interferometers as highly sensitive optical fiber platforms for bacteria sensing”. The title is more understandable and less confusing.
The authors have considered the most of my comments and suggestions and in my opinion the manuscript was improved.
I have found that some sentences are without spaces, probably due to the pdf conversion but it should be verified and corrected as: “ While 30 DTP-LPGs have been more exploredfor bacteria detection in102 – 106 CFU/mL for S. aureus and 103 - 31 109 CFU/mLfor E.coli, the IMZIs reached 102 – 108 CFU/mL for E. coli and have a potential for 32 becomingpracticalpLbacteria sensors.”
Overall, the manuscript is in conditions to be published.
Author Response
Dear Reviewer,
Thank you very much for your last comments.
Below in red are our responses.
Comments and Suggestions for Authors
I have found that some sentences are without spaces, probably due to the pdf conversion but it should be verified and corrected as: “ While 30 DTP-LPGs have been more exploredfor bacteria detection in102 – 106 CFU/mL for S. aureus and 103 - 31 109 CFU/mLfor E.coli, the IMZIs reached 102 – 108 CFU/mL for E. coli and have a potential for 32 becomingpracticalpLbacteria sensors.”
The problem seems to be a Microsoft Word problem. Such drop outs of spaces occur by themselves.
We have corrected the errors reported by you, and have found and corrected others which popped out on re-opening the files.
It appears that the final corrections will be in the proofs.

Reviewer 3 Report
Line 136: Authors claim that 'we make use of this effect to obtain maximum sensitivity...' was the work done in the authors' lab? any references to include?
Fig 2 is for RI changes. Indicate how RI changes in the figure.
Not sure if it happened when converting to .pdf, there is no space between many words. for example, line 159 'dB/RIU for RI below1.34 that isreduces to 124 dB/RIU beyond that (Fig. 3b). The IR DTP-LPG has the
160 same sensitivity in the whole RI range reachingabout 317dB/RIU (Fig. 4b)'. line 198 'where the sensor length is up toca. 50mm, ...' Many places like these.
Line 171: 'The latter, are not limited in period length, but..' specify what 'latter' refer to
Line 174: 'The biosensors reported here were fabricated using KrF excimer laser in combination with an amplitude mask and a subsequent etching.' This is a review paper. The word 'report' may not be appropriate.
This is a review paper, however, authors often mention 'we' or 'in our case' line 276. It is confusion. It might be better to state what has been done in the authors' lab and reference to previous work since the review paper intents to review the work done previously.
Author Response
Dear Reviewer,
Thank you very much for your last detailed comments and suggestions.
Our responses are given in red after your remarks.
Line 136: Authors claim that 'we make use of this effect to obtain maximum sensitivity...' was the work done in the authors' lab? any references to include?
Yes, the results from the measurements shown in Fig.2, Fig.3 and Fig.4 are on gratings fabricated at the Photonic Research Center, Canada and we have added a comment.
Fig 2 is for RI changes. Indicate how RI changes in the figure.
We have added the initial and final values of the SRI at the beginning and at the end of the arrows.
Not sure if it happened when converting to .pdf, there is no space between many words. for example, line 159 'dB/RIU for RI below1.34 that isreduces to 124 dB/RIU beyond that (Fig. 3b). The IR DTP-LPG has the
160 same sensitivity in the whole RI range reachingabout 317dB/RIU (Fig. 4b)'. line 198 'where the sensor length is up toca. 50mm, ...' Many places like these.
This seems to be a problem of the Microsoft Word. We notice the drop out of spaces after re-opening the file. We have corrected all found errors. Yet the final corrections will be at the proofs.
Line 171: 'The latter, are not limited in period length, but..' specify what 'latter' refer to
We have added the expression “Amplitute mask and point-by-point laser written LPGs” instead of “the latter”
Line 174: 'The biosensors reported here were fabricated using KrF excimer laser in combination with an amplitude mask and a subsequent etching.' This is a review paper. The word 'report' may not be appropriate.
We substitute the expression with “The described biosensors…”
This is a review paper, however, authors often mention 'we' or 'in our case' line 276. It is confusion. It might be better to state what has been done in the authors' lab and reference to previous work since the review paper intents to review the work done previously.
We have modified the sentence as “The micromachining is similar to previously reported techniques [14, 36-42] was performed using a Solstice Ti: Sapphire fs laser”.
